# Long Short-Term Memory as a Dynamically Computed Element-wise Weighted Sum

## Abstract

Long short-term memory networks (LSTMs) were introduced to combat vanishing gradients in simple recurrent neural networks (S-RNNs) by augmenting them with additive recurrent connections controlled by gates. We present an alternate view to explain the success of LSTMs: the gates themselves are powerful recurrent models that provide more representational power than previously appreciated. We do this by showing that the LSTM's gates can be decoupled from the embedded S-RNN, producing a restricted class of RNNs where the main recurrence computes an element-wise weighted sum of context-independent functions of the inputs. Experiments on a range of challenging NLP problems demonstrate that the simplified gate-based models work substantially better than S-RNNs, and often just as well as the original LSTMs, strongly suggesting that the gates are doing much more in practice than just alleviating vanishing gradients.

## 1 Introduction

Long short-term memory networks (LSTM) (Hochreiter & Schmidhuber, 1997) have become the de-facto recurrent neural network (RNN) for learning representations of sequences in many research areas, including natural language processing (NLP). Like simple recurrent neural networks (S-RNNs) (Elman, 1990), LSTMs are able to learn non-linear functions of arbitrary-length input sequences. However, they also introduce an additional memory cell to mitigate the vanishing gradient problem (Hochreiter, 1991; Bengio et al., 1994). This memory is controlled by a mechanism of gates, whose additive connections allow long-distance dependencies to be learned more easily during backpropagation. While this view is mathematically accurate, in this paper we argue that it does not provide a complete picture of why LSTMs work in practice.

We present an alternate view to explain the success of LSTMs: the gates themselves are powerful recurrent models that provide more representational power than previously appreciated. To demonstrate this, we first show that LSTMs can be seen as a combination of two recurrent models: (1) an S-RNN, and (2) an element-wise weighted sum of the S-RNN's outputs over time, which is implicitly computed by the gates. We hypothesize that, for many practical NLP problems, the weighted sum serves as the main modeling component. The S-RNN, while theoretically expressive, is in practice only a minor contributor that clouds the mathematical clarity of the model. By replacing the S-RNN with a context-*independent* function of the input, we arrive at a much more restricted class of RNNs, where the main recurrence is via the element-wise weighted sums that the gates are computing.

We test our hypothesis on NLP problems, where LSTMs are wildly popular at least in part due to their ability to model crucial language phenomena such as word order (Adi et al., 2017), syntactic structure (Linzen et al., 2016), and even long-range semantic dependencies (He et al., 2017). We consider four challenging tasks: language modeling, question answering, dependency parsing, and machine translation. Experiments show that while removing the gates from an LSTM can severely hurt performance, replacing the S-RNN with a simple linear transformation of the input results in minimal or no loss in model performance. We further show that in many cases, LSTMs can be further simplified by removing the output gate, arriving at an even more transparent architecture, where the output is a context-*independent* function of the weighted sum. Together, these results suggest that the gates' ability to compute an element-wise weighted sum, rather than the non-linear transition dynamics of S-RNNs, are the driving force behind LSTM's success.

## 2 THE MEMORY CELL COMPUTES AN ELEMENT-WISE WEIGHTED SUM

LSTMs are typically motivated as an augmentation of simple RNNs (S-RNNs), defined as follows:

$$h_t = \tanh(W_{hh}h_{t-1} + W_{hx}x_t + b_h) \tag{1}$$

S-RNNs suffer from the vanishing gradient problem (Hochreiter, 1991; Bengio et al., 1994) due to compounding multiplicative updates of the hidden state. By introducing a memory cell and an output layer that are controlled by a set of gates, LSTMs enable shortcuts through which gradients can flow easily when learning with backpropagation. This mechanism enables learning of long-distance dependencies while preserving the expressive power of recurrent non-linear transformations provided by S-RNNs.

Rather than viewing the gates as simply an auxiliary mechanism to address a *learning* problem, we present an alternate view that emphasizes their *modeling* strengths. We argue that the LSTM should be interpreted as a hybrid of two distinct recurrent architectures: (1) the S-RNN which provides multiplicative connections across timesteps, and (2) the memory cell which provides additive connections across timesteps. On top of these recurrences, an output layer is included that simply squashes and filters the memory cell at each step.

Throughout this paper, let $\{x_1, \ldots, x_n\}$ be the sequence of input vectors, $\{h_1, \ldots, h_n\}$ be the sequence of output vectors, and $\{c_1, \ldots, c_n\}$ be the memory cell's states. Then, given the basic LSTM definition below, we can formally identify three sub-components.

$$\widetilde{c}_t = \tanh(W_{ch}h_{t-1} + W_{cx}x_t + b_c) \tag{2}$$
$$i_t = \sigma(W_{ih}h_{t-1} + W_{ix}x_t + b_i) \tag{3}$$
$$f_t = \sigma(W_{fh}h_{t-1} + W_{fx}x_t + b_f) \tag{4}$$
$$c_t = i_t \circ \widetilde{c}_t + f_t \circ c_{t-1} \tag{5}$$
$$o_t = \sigma(W_{oh}h_{t-1} + W_{ox}x_t + b_o) \tag{6}$$
$$h_t = o_t \circ \tanh(c_t) \tag{7}$$

**Content Layer (Equation 2)**   We refer to $\widetilde{c}_t$ as the content layer, which is the output of an S-RNN. Evaluating the need for the multiplicative recurrent connections in this content layer is the focus of this work. The content layer is passed to the memory cell, which decides which parts of it to store.

**Memory Cell (Equations 3-5)**   The memory cell $c_t$ is controlled by two gates. The input gate $i_t$ controls what part of the content ($\widetilde{c}_t$) is written to the memory, while the forget gate $f_t$ controls what part of the memory is deleted by filtering the previous state of the memory ($c_{t-1}$). Writing to the memory is done by adding the filtered content ($i_t \circ \widetilde{c}_t$) to the retained memory ($f_t \circ c_{t-1}$).

**Output Layer (Equations 6-7)**   The output layer $h_t$ passes the memory cell through a $\tanh$ activation function and uses an output gate $o_t$ to read selectively from the squashed memory cell.

Our goal is to study how much each of these components contribute to the empirical performance of LSTMs. In particular, it is worth considering the memory cell in more detail to reveal why it could serve as a standalone powerful model of long-distance context. It is possible to show that it implicitly computes an *element-wise weighted sum* of all the previous content layers by expanding the recurrence relation in equation (5):

$$c_t = i_t \circ \widetilde{c}_t + f_t \circ c_{t-1}$$
$$= \sum_{j=0}^{t} \left( i_j \circ \prod_{k=j+1}^{t} f_k \right) \circ \widetilde{c}_j \tag{8}$$
$$= \sum_{j=0}^{t} w_j^t \circ \widetilde{c}_j$$

Each weight $w_j^t$ is a product of the input gate $i_j$ (when its respective input $\widetilde{c}_j$ was read) and every subsequent forget gate $f_k$. An interesting property of these weights is that, like the gates, they are also soft element-wise binary filters.[1]

---

[1] GRUs also exhibit the same property of computing weighted sums over a content. Specifically, they compute weighted-*averages*, since the gates are coupled.

This sum is similar to recent architectures that rely on self-attention to learn context-dependent word representations (Cheng et al., 2016; Parikh et al., 2016; Vaswani et al., 2017). There are two major differences from self-attention: (1) instead of computing a weighted sum for each attention head, a separate weighted sum is computed for every dimension of the memory cell, (2) the weighted sum is accumulated with a dynamic program, enabling a linear rather than quadratic complexity in comparison to self-attention.

## 3   MEMORY CELLS ARE POWERFUL STANDALONE MODELS

The restricted space of element-wise weighted sums allows for easier mathematical analysis, visualization, and perhaps even learnability. However, constrained function spaces are also less expressive, and a natural question is whether these models will work well for NLP problems that need highly contextualized word representations. We hypothesize that the memory cell (which computes weighted sums) can function as a standalone contextualizer. To test this hypothesis, we present several simplifications of the LSTM's architecture (Section 3.1), and show on a variety of NLP benchmarks that there is a qualitative performance difference between models that contain a memory cell and those that do not (Section 3.2). We conclude that the content and output layers are relatively minor contributors, and that the space of element-wise weighted sums is sufficiently powerful to compete with fully parameterized LSTMs (Section 3.3).

### 3.1   SIMPLIFIED MODELS

The modeling power of LSTMs is commonly assumed to derive from the S-RNN in the content layer, with the rest of the model acting as a learning aid to bypass the vanishing gradient problem. We first isolate the S-RNN by ablating the gates (denoted as *LSTM – GATES* for consistency).

To test whether the memory cell has enough modeling power of its own, we take an LSTM and replace the S-RNN in the content layer from Equation 2 with a simple linear transformation, creating the *LSTM – S-RNN* model:

$$
\begin{aligned}
\widetilde{\boldsymbol{c}}_t &= \boldsymbol{W}_{cx}\boldsymbol{x}_t \\
\boldsymbol{i}_t &= \sigma(\boldsymbol{W}_{ih}\boldsymbol{h}_{t-1} + \boldsymbol{W}_{ix}\boldsymbol{x}_t + \boldsymbol{b}_i) \\
\boldsymbol{f}_t &= \sigma(\boldsymbol{W}_{fh}\boldsymbol{h}_{t-1} + \boldsymbol{W}_{fx}\boldsymbol{x}_t + \boldsymbol{b}_f) \\
\boldsymbol{c}_t &= \boldsymbol{i}_t \circ \widetilde{\boldsymbol{c}}_t + \boldsymbol{f}_t \circ \boldsymbol{c}_{t-1} \\
\boldsymbol{o}_t &= \sigma(\boldsymbol{W}_{oh}\boldsymbol{h}_{t-1} + \boldsymbol{W}_{ox}\boldsymbol{x}_t + \boldsymbol{b}_o) \\
\boldsymbol{h}_t &= \boldsymbol{o}_t \circ \tanh(\boldsymbol{c}_t)
\end{aligned}
\tag{9}
$$

We further simplify the LSTM by removing the output gate from Equation 7, leaving only the activation function in the output layer (*LSTM – S-RNN – OUT*):

$$
\begin{aligned}
\widetilde{\boldsymbol{c}}_t &= \boldsymbol{W}_{cx}\boldsymbol{x}_t \\
\boldsymbol{i}_t &= \sigma(\boldsymbol{W}_{ih}\boldsymbol{h}_{t-1} + \boldsymbol{W}_{ix}\boldsymbol{x}_t + \boldsymbol{b}_i) \\
\boldsymbol{f}_t &= \sigma(\boldsymbol{W}_{fh}\boldsymbol{h}_{t-1} + \boldsymbol{W}_{fx}\boldsymbol{x}_t + \boldsymbol{b}_f) \\
\boldsymbol{c}_t &= \boldsymbol{i}_t \circ \widetilde{\boldsymbol{c}}_t + \boldsymbol{f}_t \circ \boldsymbol{c}_{t-1} \\
\boldsymbol{h}_t &= \tanh(\boldsymbol{c}_t)
\end{aligned}
\tag{10}
$$

After removing the S-RNN and the output gate from the LSTM, the entire ablated model can be written in a modular, compact form:

$$
\boldsymbol{h}_t = \text{OUTPUT}\Big( \sum_{j=0}^{t} \boldsymbol{w}_j^t \circ \text{CONTENT}(\boldsymbol{x}_j) \Big)
\tag{11}
$$

where the content layer CONTENT($\cdot$) and the output layer OUTPUT($\cdot$) are both context-*independent* functions, making the entire model highly constrained and interpretable. The complexity of modeling contextual information is needed only for computing the weights $\boldsymbol{w}_j^t$. As we will see in Section 3.2, both of these ablations perform on par with LSTMs on language modeling, question answering, dependency parsing, and machine translation.

There are many other models that can be expressed in the weighted-sum form (Equation 11). In this work, we focus on the closest variant of LSTM that satisfies this property; removing the S-RNN and the output gate is sufficient for the content and output functions to be context-independent. We leave more thorough investigations into the necessity of the remaining architecture as future work.

## 3.2 EXPERIMENTS

We compare model performance on four NLP tasks, with an experimental setup that is lenient towards LSTMs and harsh towards its simplifications. In each case, we use existing implementations and previously reported hyperparameter settings. Since these settings were tuned for LSTMs, any simplification that performs equally to (or better than) LSTMs under these LSTM-friendly settings provides strong evidence that the ablated component is not a contributing factor. For each task we also report the mean and standard deviation of 5 runs of the LSTM settings to demonstrate the typical variance observed due to training with different random initializations.[2] The code and settings to replicate these experiments are publicly available.[3]

### 3.2.1 LANGUAGE MODELING

We evaluate on two language modeling datasets: the Penn Treebank (PTB) (Marcus et al., 1993), and Google's billion-word benchmark (BWB) (Chelba et al., 2014). PTB contains approximately 1M tokens over a vocabulary of 10K words. We used the implementation of Zaremba et al. (2014) while replacing any invocation of LSTMs with simpler models. We tested two of their configurations: *medium*, which uses two layers of 650-dimension LSTMs, and *large*, which uses two layers of 1500-dimension LSTMs.

BWB is about a thousand times larger than PTB, and uses a more diverse vocabulary of 800K words. Using the implementation of Józefowicz et al. (2016), we tested their *LSTM-2048-512* configuration. Our experiments use exactly the same hyperparameters (dimensions, dropout, learning rates, etc) that were originally tuned for LSTMs (Józefowicz et al., 2016). Following their implementation, we project the hidden state at each time step down to 512 dimensions. Due to the enormous size of this dataset, we stopped training after 5 epochs.

Table 1 shows overall model performance. In all three cases, replacing the LSTM's content layer with a linear transformation results in small differences in perplexity. The most important result is that the small fluctuations in performance between the various gated architectures are minuscule in comparison to the enormous gap between the S-RNN ($LSTM - GATES$) and the original LSTM. This striking difference strongly supports our hypothesis that the weighted sums computed by the gates – not the S-RNN – is the recurrent model that contributes mostly strongly to the final performance.

### 3.2.2 QUESTION ANSWERING

For question answering, we use two different QA systems on the Stanford question answering dataset (SQuAD) (Rajpurkar et al., 2016): the Bidirectional Attention Flow model (BiDAF) (Seo et al., 2016) and DrQA (Chen et al., 2017). BiDAF contains 3 LSTMs, which are referred to as the phrase layer, the modeling layer, and the span end encoder. Our experiments replace each of these LSTMs with their simplified counterparts. We directly use the implementation of BiDAF from AllenNLP (Gardner et al., 2017), and all experiments reuse the existing hyperparameters that were tuned for LSTMs. Likewise, we use an open-source implementation of DrQA[4] and replace only the LSTMs, while leaving everything else intact.

Table 2 shows that all the gated models do comparably. Most importantly, ablating the S-RNN from the LSTM has a minor effect in comparison to the drop in performance when ablating the gates.

### 3.2.3 DEPENDENCY PARSING

For dependency parsing, we use the Deep Biaffine Dependency Parser (Dozat & Manning, 2016), which relies on stacked bidirectional LSTMs to learn context-sensitive word embeddings for de-

---

[2]Due to time constraints, we only include the reported LSTM results for BWB (Józefowicz et al., 2015).

[3]http://anonymous

[4]https://github.com/hitvoice/DrQA

| Configuration | Model | Perplexity |
|---|---|---|
| PTB (Medium Model) | LSTM | $83.9 \pm 0.3$ |
| | – GATES | 140.9 |
| | – S-RNN | 80.5 |
| | – S-RNN – OUT | 81.6 |
| PTB (Large Model) | LSTM | $78.8 \pm 0.2$ |
| | – GATES | 126.1 |
| | – S-RNN | 76.0 |
| | – S-RNN – OUT | 78.5 |
| BWB | LSTM (Józefowicz et al., 2016) | 47.5 |
| | – GATES | 82.2 |
| | – S-RNN | 45.4 |
| | – S-RNN – OUT | 47.9 |

Table 1: The performance of simplified LSTM architectures on language modeling benchmarks, measured by perplexity.

| System | Model | EM | F1 |
|---|---|---|---|
| BiDAF | LSTM | $67.9 \pm 0.3$ | $77.5 \pm 0.2$ |
| | – GATES | 62.9 | 73.3 |
| | – S-RNN | 68.4 | 78.2 |
| | – S-RNN – OUT | 67.4 | 77.2 |
| DrQA | LSTM | $68.8 \pm 0.2$ | $78.2 \pm 0.2$ |
| | – GATES | 56.4 | 66.5 |
| | – S-RNN | 67.7 | 77.0 |
| | – S-RNN – OUT | 67.0 | 76.2 |

Table 2: The performance of simplified LSTM architectures on the question answering benchmark, SQuAD, measured by exact match (EM) and span overlap (F1).

| Model | UAS | LAS |
|---|---|---|
| LSTM | $90.60 \pm 0.21$ | $88.05 \pm 0.33$ |
| – GATES | 87.75 | 84.61 |
| – S-RNN | 90.77 | 88.49 |
| – S-RNN – OUT | 90.70 | 88.31 |

Table 3: The performance of simplified LSTM architectures on the universal dependencies parsing benchmark, measured by unlabeled attachment score (UAS) and labeled attachment score (LAS).

termining arcs between a pair of words. We directly use their released implementation, which is evaluated on the Universal Dependencies English Web Treebank v1.3 (Silveira et al., 2014). In our experiments, we use the existing hyperparameters and only replace the LSTMs with the simplified architectures.

We observe the same pattern in the ablations for dependency parsing. The differences in performance between the gated models fall within the differences between multiple experiments with LSTMs. Consistent with ablation results from other tasks, removing the gating mechanisms causes a 3-4 point drop in performance.

### 3.2.4 MACHINE TRANSLATION

For machine translation, we used OpenNMT (Klein et al., 2017) to train English to German translation models on the multi-modal benchmarks from WMT 2016 (used in OpenNMT's readme file). We use OpenNMT's default model and hyperparameters, replacing the stacked bidirectional LSTM of its

| Model | BLEU |
|---|---|
| LSTM | 35.95 |
| – GATES | 12.22 |
| – S-RNN | 36.66 |
| – S-RNN – OUT | 36.39 |

Table 4: The performance of simplified LSTM architectures on the WMT 2016 multi-modal English to German translation benchmark, measured by BLEU.

encoder with the simplified architectures. Table 4 shows that while models containing memory cells perform more-or-less on par, removing the memory cell yields a substantial performance drop.

### 3.3 DISCUSSION

In the above experiments, we show three major ablations of the LSTM. In the S-RNN experiments (*LSTM – GATES*), we ablate the memory cell and the output layer. In the *LSTM – S-RNN* and *LSTM – S-RNN – OUT* experiments, we ablate the S-RNN. As consistent with previous literature, removing the memory cell degrades performance drastically. In contrast, removing the S-RNN makes little to no difference in the final performance, suggesting that the memory cell alone is largely responsible for the success of LSTMs in NLP. The results also confirm our hypothesis that weighted sums of context words is a powerful, yet more interpretable, model of contextual information.

## 4 WEIGHT VISUALIZATION

Given the empirical evidence that LSTMs are effectively learning weighted sums of the content layers, it is natural to investigate what weights the model learns in practice. Using the more mathematically transparent simplification of LSTMs, we can visualize the weights $\boldsymbol{w}_j^t$ that are placed on every input $j$ at every timestep $t$ (see Equation 11).

Unlike attention mechanisms, these weights are vectors rather than scalar values. Therefore, we can only provide a coarse-grained visualization of the weights by rendering their $L^2$-norm, as shown in Table 5. In the visualization, each column indicates the word represented by the weighted sum, and each row indicates the word over which the weighted sum is computed. Dark horizontal streaks indicate the duration for which a word was remembered. Unsurprisingly, the weights on the diagonal are always the largest since it indicates the weight of the current word. More interesting task-specific patterns emerge when inspecting the off-diagonals that represent the weight on the context words.

The first visualization uses the language model from BWB. Due to the language modeling setup, there are only non-zero weights on the current or previous words. We find that the common function words are quickly forgotten, while infrequent words that signal the topic are remembered over very long distances.

The second visualization uses the dependency parser. In this setting, since the recurrent architectures are bidirectional, there are non-zero weights on all words in the sentence. The top-right triangle indicates weights from the forward direction, and the bottom-left triangle indicates from the backward direction. For syntax, we see a significantly different pattern. Function words that are useful for determining syntax are more likely to be remembered. Weights on head words are also likely to persist until the end of a constituent.

This illustration provides only a glimpse into what the model is capturing, and perhaps future, more detailed visualizations that take the individual dimensions into account can provide further insight into what LSTMs are learning in practice.

## 5 RELATED WORK

Many variants of LSTMs (Hochreiter & Schmidhuber, 1997) have been previously explored. These typically consist of a different parameterization the gates, such as LSTMs with peephole connections (Gers & Schmidhuber, 2000), or a rewiring of the connections, such as GRUs (Cho et al., 2014).

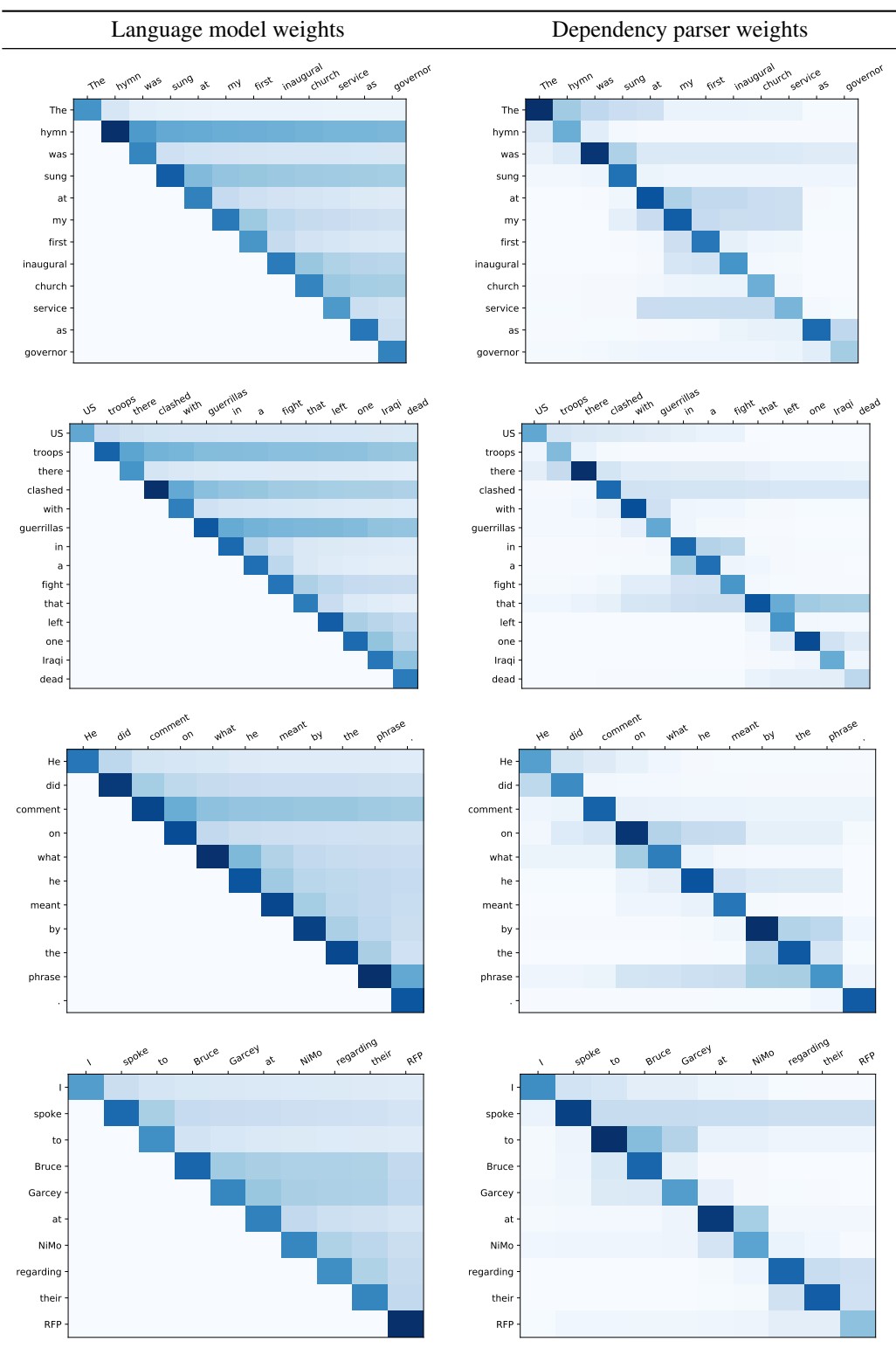

Table 5: Visualization of the weights on context words learned by the memory cell. Each column represents the current word $t$, and each row represents a context word $j$. The gating mechanism implicitly computes element-wise weighted sums over each column. The darkness of each square indicates the $L^2$-norm of the vector weights $\boldsymbol{w}_j^t$ from Equation 11. Figures on the left show weights learned by a language model. Figures on the right show weights learned by a dependency parser.

However, these modifications invariably maintain the recurrent content layer. Even more systematic explorations of LSTM variants (Józefowicz et al., 2015; Greff et al., 2016; Zoph & Le, 2017) do not question the importance of the embedded S-RNN. This is the first study to provide apples-to-apples comparisons between LSTMs and LSTMs *without* the recurrent content layer.

Several other recent works have also reported promising results with recurrent models that are vastly simpler than LSTMs, such as quasi-recurrent neural networks (Bradbury et al., 2016), strongly-typed recurrent neural networks (Balduzzi & Ghifary, 2016), kernel neural networks (Lei et al., 2017), and simple recurrent units (Lei & Zhang, 2017), making it increasingly apparent that LSTMs are over-parameterized. While these works indicate an obvious trend, their focus is not to provide insight into what exactly LSTMs are learning. In our carefully controlled ablation studies, we propose and evaluate the minimal changes required to test our hypothesis that LSTMs are powerful because they dynamically compute element-wise weighted sums of content layers.

As mentioned in Section 2, this weighted-sum view of LSTMs is highly related to neural attention (Bahdanau et al., 2015), which assigns a normalized scalar weight to each element as a function of its compatibility with an external element. The ability to inspect attention weights has driven the use of more interpretable neural models. Self-attention (Cheng et al., 2016; Parikh et al., 2016) extends this notion by computing intra-sequence attention. Vaswani et al. (2017) further showed that state-of-the-art machine translation can be achieved using only self-attention and without LSTMs. Recently, Arora et al. (2017) proposed a theory-driven approach to assign scalar weights to elements in a bag of words. The success of self-attention corroborates our findings that weighted sums are indeed a more effective method of learning context-sensitive representations than previously appreciated.

## 6 CONCLUSION

We presented an alternate view of LSTMs: they are a hybrid of S-RNNs and a gated model that dynamically computes weighted sums of the S-RNN outputs. Our experiments investigated whether the S-RNN is a necessary component of LSTMs. In other words, are the gates alone as powerful of a model as an LSTM? Results across four major NLP tasks (language modeling, question answering, dependency parsing, and machine translation) indicate that LSTMs suffer little to no performance loss when removing the S-RNN, but removing the gates can degrade performance substantially. This provides evidence that the gating mechanism is doing the heavy lifting in modeling context, and that element-wise weighted sums of context-independent functions of the inputs are often as effective as fully-parameterized LSTMs.

This work sheds light on the inner workings of the relatively opaque LSTM. By removing the S-RNN and the output gate, we also show that the resulting model is a far more mathematically transparent variant of LSTMs. This transparency enables a visualization of how the context affects the output of the model at every timestep, much like in attention-based models. We hope that this new outlook on LSTMs will foster better and more efficient models of contextualization.

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
