# OpenReview forum: "Long Short-Term Memory as a Dynamically Computed Element-wise Weighted Sum"
_ICLR.cc/2018/Conference — Reject_

### Official Review · AnonReviewer3 · 2017-11-27

**Rating:** 6
**Confidence:** 4

**Review:**

This paper proposes a simplified LSTM variants by removing the non-linearity of content item and output gate. It shows comparable results with standard LSTM.

I believe this is a updated version of https://arxiv.org/abs/1705.07393 (Recurrent Additive Networks) with stronger experimental results.

However, the formulation is very similar to "[1] Semi-supervised Question Retrieval with Gated Convolutions" 2016 by Lei, and "Deriving Neural Architectures from Sequence and Graph Kernels" which give theoretical view from string kernel about why this type of networks works. Both of the two paper don't have output gate and non-linearity of "Wx_t" and results on PTB also stronger than this paper. It also have some visualization about how the model decay the weights. Other AnonReviewer also point out some similar work. I won't repeat it here. In the paper, the author argued "we propose and evaluate the minimal changes..." but I think the these type of analysis also been covered by [1], Figure 5.

On the experimental side, to draw the conclusion, "weighted sum" is enough for LSTM. I think at least Machine Translation and other classification results should be added. I'm not very familiar with SQuAD dataset, but the results seems worse than "Reading Wikipedia to answer open-domain questions" Table 4 which seems use a vanilla LSTM setup.

Update: the revised version of the paper addresses all my concerns about experiments. So I increased my score.

---

> ### Author Response · Authors · 2017-12-03
> **Author Response**
>
> Thank you for your comments.
>
> We would like to draw the reviewer's attention to two major points:
>
> =Paper Contribution=
> The goal of this paper is not to propose a model that outperforms LSTMs, but to understand how LSTMs work. The aptly-named "LSTM - RNN" is the minimal change from the original LSTM that allows us to test whether the weighted summing mechanism (as computed by LSTMs) is sufficiently powerful to facilitate NLP models. The main point of this paper is that the gating mechanism (which is able to compute element-wise weighted sums) is a powerful model, and that the embedded vanilla RNN is redundant - at least on the variety of tasks on which we evaluated it on.
>
> =Controlled Experiments=
> By controlling every variable (model, data, hyperparameters, etc) and running head-to-head comparisons between LSTM and LSTM - RNN, we deduce whether the recurrence in the content layer is redundant. These experiments are vastly different from the ones in "Deriving Neural Architectures from Sequence and Graph Kernels", since comparing the PTB results of Lei et al with ours does not control for a wide variety of changes in the experiment. The results of Lei et al are significantly stronger due to a deviation in both model and hyperparameters from the original setting of Zaremba et al (2014), which we followed. Specifically, these deviations include:
> 1) highway connections
> 2) tying input and output embeddings
> 3) variational dropout
> 4) different dropout rates
> 5) different number of layers
> 6) different number of dimensions
> 7) different initialization scheme
> All of these factors have a massive effect on performance, vastly outweighing the performance difference between one recurrent architecture and another on benchmarks such as PTB. In particular, the combination of highway connections and additional layers accounts for about 20 perplexity points.
>
> The reason we are so familiar with these details is because we reproduced Lei et al's result, and tried to decouple the core model from these other variables. However, when we ran their model in the original Zaremba et al setting, their model did exhibit a drop in performance with respect to LSTMs. We are happy to add an additional section to the paper that demonstrates how tampering with the gating mechanism can actually reduce performance in some cases. This will complement the current experiments, which deal with simplifying the content layer.
>
> =Other Comments=
> - You compare our results on SQuAD, which are based on the BiDAF model (Seo et al, 2017), to the results of DrQA (Chen et al, 2017). These are two different QA models, and the comparison is flawed in the same way that the comparison of our PTB result to Lei et al's is invalid.
> - We are happy to run on additional benchmarks if there is a good reason to suspect that the current 4 benchmarks do not provide sufficient coverage of interesting linguistic phenomena. Specifically, the classification tasks are a poor test of RNN's modeling power since in many cases the best methods are simple bag-of-words classifiers.

---

> > ### Comment · AnonReviewer3 · 2017-12-07
> >
> > Thank you for your quick response! I'll take a closer look of the equation and experimental part and I'll update the comments / score after that. Since the author may need some time to add more results, I just leave early comments here.
> >
> > Two major concerns:
> >
> > (1) Model side, I feel the take-away message is similar to the model I pointed out. The difference is that one link to the kernel method and one is try to simplify LSTM (or make LSTM more interpretable). This is why it's not too surprised to me the "Element-wise Weighted Sum" is powerful.  This is why I feel part of conclusion is over-claimed, e.g. "This work sheds light on the inner workings of the relatively opaque LSTM", and "This transparency enables a visualization of how the context affects" (simliar graph can be seen in the paper I pointed out). But overall, it is very useful analysis for LSTM on NLP.
> >
> > (2) Experiments: I think no need to re-do all the variants for PTB. I also agree with that classification task is a poor test of RNN's modeling in some sense. But any reason not include machine translation results?
> >
> > Some questions:
> > I'm not familiar with the SQuaD data. But seems DrQA got better results than BiDAF. Any reason prefer BiDAF over DrQA?
> >
> > Did the author find any hyper-parameters difference for this simplified LSTM and a regular LSTM? Since it's remove the non-linearity of memory cell (f_t * c_t). Does it need a truncate the cell output or a more aggressive clipping?

---

> > > ### Author Response · Authors · 2017-12-07
> > > **More results on the way**
> > >
> > > Thank you for taking our response into consideration.
> > >
> > > We did not have time to run machine translation experiments, but we will run them now and update as they arrive.
> > >
> > > Regarding DrQA vs BiDAF - there is no specific preference for one model over the other except for engineering and experiment-time overhead. We will also run DrQA experiments and report the results.
> > >
> > > As for the hyperparameters - we did not modify *any* hyperparameters, so we cannot say anything about the behavior of this space with confidence.

---

### Official Review · AnonReviewer1 · 2017-11-27
**Useful anaylsis of LSTMs**

**Rating:** 5
**Confidence:** 5

**Review:**

This paper presents an analysis of LSTMS showing that they have a from where the memory cell contents at each step is a weighted combination of the “content update” values computed at each time step. The weightings are defined in terms of an exponential decay on each dimension at each time step (given by the forget gate), which lets the cell be computed sequentially in linear time rather than in the exhaustive quadratic time that would apparently be necessary for this definition. Second, the paper offers a simplification of LSTMs that compute the value by which the memory cell at each time step in terms of a deterministic function of the input rather than a function of the input and the current context. This reduced form of the LSTM is shown to perform comparably to “full” LSTMs.

The decomposition of the LSTM in terms of these weights is useful, and suggests new strategies for comparing existing quadratic time attention-based extensions to RNNs. The proposed model variations (which replaces the “content update” that has a recurrent network in terms of context-independent update) and their evaluations seem rather more arbitrary. First, there are two RNNs present in the LSTM- one controls the gates, one controls the content update. You get rid of one, not the other. You can make an argument for why the one that was ablated was “more interesting”, but really this is an obvious empirical question that should be addressed. The second problem of what tasks to evaluate on is a general problem with comparing RNNs. One non-language task (e.g., some RL agent with an LSTM, or learning to execute or something) and one synthetic task (copying or something) might be sensible. Although I don’t think this is the responsibility of this paper (although something that should be considered).

Finally, there are many further simplifications of LSTMs that could have been explored in the literature: coupled input-forget gates (Greff et al, 2015), diagonal matrices for gates, GRUs. When proposing yet another simplification, some sense for how these different reductions is useful, so I would recommend comparison to those.

Notes on clarity:
Before Eq 1 it’s hard to know what the antecedent of “which” is without reading ahead.

For componentwise multiplication, you have been using \circ, but then for the iterated component wise product, \prod is used. To be consistent, notation like \odot and \bigodot might be a bit clearer.

The discussion of dynamic programming: the dynamic program is also only available because the attention pattern is limited in a way that self attention is not. This might be worth mentioning.

When presenting Eq 11, the definition of w_j^t elides a lot of complexity. Indeed, w_j^t is only ever implicitly defined in Eq 8, whereas things like the input and forget gates are defined multiple times in the text. Since w_j^t can be defined iteratively and recursively (as a dynamic program), it’s probably worth writing both out, for expository clarity.

Eq 11 might be clearer if you show that Eq 8 can also be rewritten in the same wheat, provided, you make h_{t-1} an argument to output and content.

Table 4 is unclear. In a language model, the figure looks like it is attending to the word that is being generated, which is clearly not what you want to convey since language models don’t condition on the word they are predicting. Presumably the strong diagonal attention is attending to the previous word when computing the representation to generate the subsequent word? In any case, this figure should be corrected to reflect this. This objection also concerns the right hand figure, and the semantics of the meaning of the upper vs lower triangles should be clarified in the caption (rather than just in the text).

---

> ### Author Response · Authors · 2017-12-03
> **Author Response**
>
> Thank you for your comments. We will amend the paper to address all the clarity issues you pointed out.
>
> Indeed, the gates can also be seen as vanilla RNNs, but there is also evidence that removing the recurrent nature of the gates still does not cripple an LSTM (see, for example, QRNNs https://arxiv.org/pdf/1611.01576.pdf and SRUs https://arxiv.org/pdf/1709.02755.pdf). However, both QRNNs and SRUs add alternative mechanisms that are not present in the original LSTMs; QRNNs add multi-token convolutions, while SRUs add highway connections.
>
> The experiments in our paper are the first to explicitly isolate the gating mechanism from the embedded vanilla RNN in the content layer. The model we describe as "LSTM - RNN" is the minimal change that allows us to test whether the weighted summing mechanism (as computed by LSTMs) is sufficiently powerful to facilitate NLP models.
>
> This ablation is very different from the ones in (Greff et al, 2015) and GRUs, since those models retain the recurrent content layer. We also ran similar experiments based on these models (e.g. "GRU - RNN"), but removed them from the paper to keep the discussion focused and succinct. We are happy to expand the paper accordingly if necessary. (Also, note that both GRUs and gate-coupled LSTMs are computing weighted averages rather than weighted sums, which appears to yield slightly lower performance in some tasks from preliminary experiments.)
>
> Regarding evaluation on a non-NLP task: our goal was to get a better understanding of why LSTMs are so useful for NLP. We will make sure that our claims are hedged accordingly. We are also happy to run experiments on an additional non-language task if this is a major concern.

---

### Official Review · AnonReviewer2 · 2017-11-27

**Rating:** 7
**Confidence:** 3

**Review:**

Summary: the paper proposes a new insight to LSTM in which the core is an element-wise weighted sum. The paper then argues that LSTM is redundant by keeping only input and forget gates to compute the weights. Experimental results show that the simplified versions work as well as the full LSTM.


Comment: I kinda like the idea and welcome this line of research. The paper is very well written and has nice visualisation of demonstrating weights. I have only one question:

in the simplified versions, content(x_t) = Wx_t , which works very well (outperforming full LSTM). I was wondering if the problem is from the tanh activation function (eq 2). What if content(x_t) = W_1 . h_{t-1} + W_2 . x_t?

---

> ### Author Response · Authors · 2017-12-03
> **Author Response**
>
> Thank you for your comments.
>
> Regarding the question about tanh, we did run preliminary experiments with the suggested setting, in which the model is identical to an LSTM save for the absence of a tanh in the content layer. Performance was very similar to the original LSTM. We also experimented with a content layer of tanh(Wx), and found the results to be very similar to those of "LSTM - RNN" that were presented in the paper.

---

### Public Comment · (anonymous) · 2017-10-30
**Related work**

A paper from ICLR last year introduced your decomposition in equation (8) (their equation (9) looks the same), and interpreted the weights as importance scores, in a manner similar to your table 4. How does this characterization of LSTMs differ from your characterization of them as "dynamically computed element-wise weighted sum"?

https://arxiv.org/abs/1702.02540

---

> ### Author Response · Authors · 2017-10-31
> **Re: Related work**
>
> Thank you for the reference! Your idea of using a function of the gates to compute importance scores is very relevant, and we will include a citation in future revisions. However, our focus is on the understanding and model alternatives that come from ablating different model internals (e.g. the S-RNN). We hope you will agree that this is very different from (but hopefully complementary to) your focus on rule extraction. Look forward to reading your paper more carefully soon!

---

### Public Comment · ~Jared_Ostmeyer1 · 2017-10-31
**Deepened my understanding of LSTM models**

I begrudgingly admit that I really like this paper! It describes the LSTM model in a new light, helping me to better understand the nuts and bolts of how it works. I think the author(s) truly understand the LSTM model.

I say begrudgingly because I fell sad they did not cite my work on recurrent weighted averages (https://arxiv.org/abs/1703.01253) or this work on recurrent discounted attention units (https://arxiv.org/abs/1705.08480). That said, the field is moving fast and it hard to both stay on top of the literature and to find space to cite every little paper.

---

> ### Author Response · Authors · 2017-10-31
> **Re: Deepened my understanding of LSTM models**
>
> Thanks for the feedback! We are happy to add more citations, sorry for missing yours. We hope our work is relevant to a wide range of existing and future work on designing model variants. The fact that there is so much work shows how important it is to understand more precisely what LSTMs are computing in practice.

---

### Public Comment · (anonymous) · 2017-11-01
**Significant overlapping content to another ICLR 2018 paper**

It is surprising to see that another earlier submitted paper titled "Dependent bidirectional recurrent neural network with super-long short-term memory" has done almost the same work in their second section for LSTM analysis.

---

> ### Author Response · Authors · 2017-11-02
> **Re: Significant overlapping content to another ICLR 2018 paper**
>
> Thank you for the pointing out this simultaneous and related work! Section 2 is not the novel part of our paper; it is a background section that presents a review of LSTMs and some simple algebra to highlight a known expansion of the memory cell's equation. As pointed out below, our contribution is in the later sections, which highlight the understanding and model alternatives that come from ablating different LSTM model internals (e.g. the S-RNN). However, it is interesting to see different papers observing the same simple recurrence, and taking it in very different directions.

---

### Comment · AnonReviewer2 · 2017-11-20
**proper evaluation?**

Although I welcome this work, I'm afraid that the authors jumped to the conclusion "that the content and output layers are redundant, and that the space of element-wise weighted sums is sufficiently powerful to compete with fully parameterized LSTMs" too quickly.

All comparison against LSTMs (and of course, any other architectures) should be done carefully, especially when there is evidence that LSTMs weren't properly tuned. For instance, Melis et al. (https://arxiv.org/pdf/1707.05589.pdf) point out that LSTMs if tuned properly can outperform most state-of-the-art models. In Melis et al. LSTMs achieve 59.6 on PTB, whereas the authors' LSTMs reach 78.8, which is much much far behind.

For dependency parsing, I was wondering why the authors didn't do the comparison on Penn Treebank, which is more standard and reported by Dozat & Manning, 2016.

---

> ### Author Response · Authors · 2017-11-21
> **Re: proper evaluation?**
>
> Of course, we agree that hyperparameters can have a dramatic effect on performance, which is why we chose readily-available systems that were tuned for LSTMs as our baselines, and did not modify any of these hyperparameters when trying out our alternatives. This puts the simplified models under a "devil's advocate" benchmark, since they could potentially benefit from other hyperparameter settings, but are forced to run with hyperparameters that were tuned for LSTMs.
>
> For PTB language modeling, we chose Zaremba et al's setting [1] because it is widely used as a baseline by other work (see, for example, Gal & Ghahramani [2] and Press & Wolf [3]) and their code is publicly available. We did not use Melis et al's setting [4] because:
> A) their code is not publicly available
> B) the complete set of optimal hyperparameters is not explicitly mentioned in the paper
> C) the paper has yet to pass peer review
> D) we already had experimental results for PTB before the paper appeared on arXiv
>
> As for dependency parsing, the code of Dozat and Manning [5] runs by default on universal dependencies (UD), which is publicly available (unlike PTB, which is proprietary). UD is also a significantly larger benchmark than PTB, and it was originally annotated as dependencies.
> See: https://github.com/tdozat/Parser-v1 under "How do you run the model?"
>
> We note that, in addition to these two benchmarks, we also tested on a much larger language-modeling dataset (Google's Billion Word Benchmark [6]) and question answering (SQuAD [7]). Our results were consistent across all four benchmarks.
>
> [1] https://arxiv.org/pdf/1409.2329.pdf
> [2] https://arxiv.org/pdf/1512.05287.pdf
> [3] https://arxiv.org/pdf/1608.05859.pdf
> [4] https://arxiv.org/pdf/1707.05589.pdf
> [5] https://arxiv.org/pdf/1611.01734.pdf
> [6] https://arxiv.org/pdf/1602.02410.pdf
> [7] https://arxiv.org/pdf/1606.05250.pdf

---

### Decision · Program_Chairs · 2018-01-29
**ICLR 2018 Conference Acceptance Decision**

**Decision:**

Reject

**Comment:**

The paper performs an ablation analysis on LSTM, showing that the gating component is the most important. There is little novelty in the analysis, and in its current form, its impact is rather limited.